# Screening of differentially expressed immune-related genes from spleen of broilers fed with probiotic *Bacillus cereus* PAS38 based on suppression subtractive hybridization

Jiajun Li[1☯], Wanqiang Li[1☯], Jianzhen Li[1,2☯], Zhenhua Wang[2], Dan Xiao[1], Yufei Wang[1], Xueqin Ni[1], Dong Zeng[1], Dongmei Zhang[1], Bo Jing[1], Lei Liu[1], Qihui Luo[1], Kangcheng Pan[1]*

1 College of Veterinary Medicine, Sichuan Agricultural University, Chengdu, Sichuan Province, China, 2 Branch of Animal Husbandry and Veterinary Medicine, Chengdu Vocational College of Agricultural Science and Technology, Chengdu, Sichuan Province, China

☯ These authors contributed equally to this work.
* pankangcheng71@126.com

**Data Availability Statement:** All ESTs sequences are available from the NCBI's dbEST database

## Abstract

The aim of this study was to construct the spleen differential genes library of broilers fed with probiotic *Bacillus cereus* PAS38 by suppression subtractive hybridization (SSH) and screen the immune-related genes. Sixty seven-day-old broilers were randomly divided into two groups. The control group was fed with basal diet, and the treated group was fed with basal diet containing *Bacillus cereus* PAS38 1×10⁶ CFU/g. Spleen tissues were taken and extracted its total RNA at 42 days old, then SSH was used to construct differential gene library and screen immune-related genes. A total of 119 differentially expressed sequence tags (ESTs) were isolated by SSH and 9 immune-related genes were screened out by Gene ontology analysis. Nine differentially expressed genes were identified by qRT-PCR. *JCHAIN*, *FTH1*, *P2RX7*, *TLR7*, *IGF1R*, *SMAD7*, and *SLC7A6* were found to be significantly up-regulated in the treated group. Which was consistent with the results of SSH. These findings imply that probiotic *Bacillus cereus* PAS38-induced differentially expressed genes in spleen might play an important role in the improvement of immunity for broilers, which provided useful information for further understanding of the molecular mechanism of probiotics responsible to affect the poultry immunity.

## Introduction

*Bacillus spp*. is a common probiotic and widely used in poultry industry [1]. A large number of studies have shown that *Bacillus* could promote the development of immune organs and activate immune-related signaling pathways serving as an immune activator, thereby stimulating the specific and non-specific immunity of the host and improving the immune ability of animals [2–4]. The research on the mechanism of *Bacillus* immune action, especially the

(GenBank accession numbers JZ980559-JZ980677.).

**Funding:** This work was supported by Natural Science Foundation of China awarded to K.C.P. (Grant number 31472130), and by the Program for Changjiang Scholars and Innovative Research Team in University awarded to X.Q.N. (Grant number IRT0848), and by the Projects of Science and Technology Innovation Research Team in the University of Sichuan Province awarded to K.C.P. (Grant number KM406183.1). The funders had no role in study design, data collection and analysis, decision to publish, or preparation of the manuscript.

**Competing interests:** The authors have declared that no competing interests exist.

molecular mechanism, is a hot spot at present. Previous papers showed that *Bacillus subtilis* improved the animal immunity by increasing the expression of cytokine genes such as *IL-2*, *IL-4*, *IL-10* and *TNF-α* in cecum and ileum mucosa of broilers [5]. And it was found that *Bacillus subtilis* could promote the expression of tight junction protein JAM-2, mucin, ZO-1 and other zonula occludens genes in intestinal mucosa of broilers [6]. These results preliminarily explained the work mechanism of probiotics from the perspective of intestinal mucosal immunity.

*Bacillus cereus* is a common soil bacterium. Some of its strains have been proved to be probiotics and have been developed as probiotics and applied to animal husbandry and veterinary fields [7]. Zhao et al. added $10^7$ CFU/g *Bacillus cereus* EN25 to the diet of sea cucumber, and found that it could significantly improve the immune function of sea cucumber and reduce the cumulative mortality rate after *Vibrio splenovirus* infection [8]. Scharek et al. added *Bacillus cereus* var. toyoi to the feed of sows and piglets, it was found that both jejunal epithelium and Peyer's patch CD8 + T cells and γδ T cells increased significantly, and the frequency of pathogenic *Escherichia coli* in piglet feces of probiotic group was also lower, which indicates that it was beneficial to the health of piglets [9]. Pan et al. found that adding 0.1% *Bacillus cereus* PAS38 to the basic diet of weanling male rabbits could reduce the number of somatostatin (SS) positive cells and the expression intensity of SS cells in the small intestine of rabbits, increase the number of 5-hydroxytryptamine (5-HT) immunoreactive cells in the small intestine and increase the expression intensity of 5-HT cells, and the combination of *Bacillus cereus* PAS38 and β-mannanase is more effective [10]. These studies indicate that some beneficial strains of *Bacillus cereus* have great potential in improving animal immunity.

SSH is a technique for constructing differential gene library, and it has many advantages, such as high sensitivity, rapidity, simplicity, low false positive rate and enrichment of rare transcripts [11]. Therefore, it is widely used in screening of differentially expressed genes in animals [12]. SSH library had been used to analyze the different hair phenotype (curly and noncurly) of Chinese Tan sheep at different growth stages [13], 67 differentially expressed genes were found, and further study confirmed that *KRT71* gene was highly correlated with the curly hair phenotype of Tan sheep. In the study of the molecular mechanism of ovarian development in Yellow River carp [14], 78 differentially expressed genes were obtained by comparing the second stage ovaries and mature ovaries of Yellow River carp with SSH. Furthermore, it was found that 78 differentially expressed genes were mainly involved in signal transduction, protein hydrolysis, cell differentiation, TGF-β signaling pathway and other biological reactions. In addition, it was also confirmed that *BMP2B*, *DESMIN* and *FP1* genes might be the biomarkers for early ovarian development.

Previous studies on the improvement of animal immunity by probiotic *Bacillus* mainly focused on its effects on the development of immune organs and the expression of serum cytokines. The screening of differentially expressed genes related to immunity in spleen tissue transcriptome after feeding broilers with probiotic *Bacillus cereus* has not been reported. Thus, it is necessary to carry out relevant experiments to further study the work mechanism of probiotics.

In this experiment, broilers were fed with *Bacillus cereus* PAS38, spleens were collected and total RNA was extracted at 42 days old. Then differential expression genes library was constructed by SSH, and differential expression immune-related genes were screened. Absolute qRT-PCR was used to verify the differential genes, in order to explore the key genes regulating immunity of *Bacillus cereus* PAS38, and lay a foundation for wide application in broilers production.

## Materials and methods

### Ethics statement

All animal experiments were performed in accordance with the guidelines for the care and use of laboratory animals and approved by the Institutional Animal Care and Use Committee of Sichuan Agricultural University (approval number: DYS20174513-1).

### Laboratory animals and strains

Sixty one-day-old avian white feather broilers were purchased from Sichuan Zhengda Animal and Poultry Co., Ltd. *Bacillus cereus* PAS38 strain was provided by Animal Microecology Institute of Sichuan Agricultural University. *Bacillus cereus* PAS38 preparation was obtained by solid fermentation (S1 Table) at 37°C for 48 h, collecting spores, drying at 65°C for 2 h and grinding. The number of viable spores was counted by plate dilution coating method and the number of viable spores was adjusted to $10^9$ CFU/g by adding corn flour.

### Reagents and instruments

The reagents used in this experiment mainly included RNAiso Plus (Takara, JPN), Smarter™ PCR cDNA Synthesis Kit (Takara, JPN), Advantage 2 PCR Kit (Takara, JPN), PCR Select™ cDNA Subtraction Kit (Takara, JPN), pUCm T vector (Sangon Biotech, CHN), DH5α Chemically Competent Cell (Beijing TsingKe, CHN), and iTaq Universl SYBR Green supermix (Bio-Rad, USA). The main instruments used were Microvolume Spectrophotometers Nano Drop 2000 (Thermo, USA), Real-time PCR thermal cycle instrument Bio Rad CFX 96 (Bio-Rad, USA), Thermostatic water bath oscillators SHZ A (Shanghai Boxun, CHN), Constant temperature incubator DHP 9080B (Shanghai Langgan, CHN), Gel imaging system Gel Doc™ XR+ (Bio-Rad, USA), High speed freezing centrifuge Micro 21R (Thermo, USA), PCR instrument ABI Veriti 96 Well (Thermo, USA), and Horizontal Electrophoresis System Mini Sub cell GT (Bio-Rad, USA).

### Experimental design

Sixty one-day-old avian white feather broilers were pre-fed with basic diet for 7 days to stabilize the metabolic conditions. The broilers were randomly divided into control group and treated group, each group consisted of three replicates with 10 chickens per replicate. The control group was fed the basal diet (Table 1). The treated group was fed the basal diet supplemented with 0.1% *Bacillus cereus* PAS38 preparation, and including *Bacillus cereus* $1 \times 10^6$ CFU per gram of feed. The experimental chickens were raised in single-layer cage with ten broilers in each cage. Each cage is equipped with two food troughs and one water trough. The control group and the treated group were fed to two animal houses respectively to avoid spore interference. Broiler chickens were fed the diet and water *ad libitum*. The chicken house was cleaned every morning and evening, and the air circulation in the chicken house was maintained. The insulation lamp was used to keep the temperature in the room at about 25°C. At the age of 42 days, two broilers were randomly selected from each repeat of each group, that was, six broilers in each group were executed and the spleens were taken and immediately stored in liquid nitrogen.

### RNA extraction and construction of cDNA libraries

Spleen tissues were ground rapidly in liquid nitrogen. Total RNA was extracted by RNAiso Plus reagent and dissolved in RNase-free deionized water. And six tubes of total RNA in each group were merged into one tube. The purity and concentration of the RNA were determined

**Table 1. Raw material composition and nutritional level of basic dietary (air-dry basis).**

| Ingredients(%) | | | | Nutrition levels | | |
|---|---|---|---|---|---|---|
| | 1d-21d | 22d-42d | | | 1d-21d | 22d-42d |
| Corn | 61.20 | 65.20 | Metabolizable energy (MJ/kg) | | 12.54 | 12.80 |
| Soybean meal | 23.00 | 18.00 | Crude protein | | 20.70 | 19.00 |
| Extruded soybean | 8.50 | 10.00 | Lysine | | 1.12 | 0.96 |
| Import fish meal | 3.00 | 3.00 | Methionine | | 0.53 | 0.43 |
| $CaHPO_4$ | 1.60 | 1.40 | Calcium | | 0.99 | 0.89 |
| Limestone | 1.10 | 1.00 | Available phosphorus | | 0.51 | 0.46 |
| NaCl | 0.32 | 0.30 | | | | |
| DL- methionine | 0.18 | 0.10 | | | | |
| L- lysine | 0.10 | | | | | |
| Premix | 1.00 | 1.00 | | | | |
| Total | 100.00 | 100.00 | | | | |

Premix is provided for feed per kg: VD3 200 IU, VA 1500 IU, VE 10 IU, VK 0.5 mg, VB12 0.01 mg, VB6 3.0 mg, VB1 1.5 mg, Nicotinic acid 30 mg, D-pantothenic acid 10 mg, Folic acid 0.5 mg, Biotin 0.15 mg, Trace elements Cu, Fe, Zn, Mn, Se, I are 8 mg, 80 mg, 40 mg, 60 mg, 0.15 mg, 0.18 mg respectively. Metabolic energy was calculated and the rest was measured.

by Nucleic acid analyzer NanoDrop 2000. Then the single-stranded cDNA was synthesized immediately by Smarter PCR cDNA Synthesis Kit (Clontech, USA). The synthetic system was as follows: 1.0μg total RNA, 1.0μL Smarter 3'-CDS Primer II A, 1.5μL RNase-free deionized water, 4.5μL total volume. The mixtures were incubated at 72°C for 3 min, and then incubated at 42°C for 2 min. Next, added 2.0μL 5×First-Strand buffer, 0.25μL DTT, 1.0μL dNTP mix, 0.25μL RNase inhibitor, 1μL Smarter II A oligo, 1.0μL Smart scribe reverse transcriptase to the mixtures, and they were incubated at 42°C for 1.5 h. The synthesized single-stranded cDNA was diluted 5-fold with TE buffer, and then stored at—20°C refrigerator.

Advantage 2 PCR Kit (Clontech, USA) was used for the synthesis of double-stranded cDNA. The reaction system was as follows: the 1.5μL single-stranded cDNA was added with 8.5μL deionized water to the total volume of 10μL, and added 10μL 10×Advantage 2 PCR buffer, 2μL 50×dNTP mix, 2μL 5'-PCR Primer II A, 2μL 50×Advantage 2 Polymerase mix, 74μL deionized water. The mixtures were pre-denatured at 95°C for 1 min, denatured at 95°C for 15 s, annealed at 65°C for 30 s, extended at 68°C for 6 min, totally 30 cycles. From the 18th cycle to the 30th cycle, the 5μL mixture was taken every three cycles for 1.2% agarose gel electrophoresis to select the optimal cycle number for constructing the cDNA libraries.

## Construction of SSH libraries

The double-stranded cDNA with the optimal cycle number was taken and digested with Rsa I enzyme at 37°C for 3 hours. Then the forward and reverse SSH differential gene libraries were constructed by Smarter PCR cDNA Synthesis Kit (Clontech, USA). In the forward library, the spleen cDNA of broilers in the treated group was used as tester, and the spleen cDNA of broilers in the control group was used as driver for hybridization. On the contrary, in the reverse library, the spleen cDNA of broilers in the control group was used as tester, and the spleen cDNA of broilers in the treated group was used as driver for hybridization.

The first hybridization mixtures were incubated at 98°C for 1.5 min and then at 68°C for 10 h. Then the driver cDNA was immediately added to the mixtures, and they were incubated at 68°C for 10 h. After the two rounds of hybridization reaction, the hybridization products were diluted with 200μL dilution buffer. Then the first nested PCR was performed using PCR

Primer 1 (primer sequence: 5'–CTAATACGACTCACTATAGGC–3') as primer. The reaction system was as follows: 1μL hybridization products, 1μL PCR Primer 1, 0.5μL 50×dNTP mix, 0.5μL 50×Advantage cDNA Polymerase mix, 2.5μL 10×Advantage 2 PCR buffer, 19.5μL deionized water. The mixtures were incubated at 75˚C for 5 min, pre-denatured at 94˚C for 30 s, denatured at 94˚C for 30 s, annealed at 66˚C for 30 s, extended at 72˚C for 2 min, totally 27 cycles.

The first Nested PCR products were diluted 10-fold with deionized water for the second Nested PCR. The reaction system was as follows: 1μL first Nested PCR products, 1μL Nested PCR primer 1 (primer sequence: 5'–TCGAGCGGCCGCCCGGGCAGGT–3'), 1μL Nested PCR primer 2R (primer sequence: 5'–AGCGTGGTCGCGGCCGAGGT–3'), 0.5μL 50×dNTP mix, 0.5μL 50×Advantage cDNA Polymerase mix,2.5μL 10×Advantage 2 PCR buffer, 18.5μL deionized water. The mixtures were pre-denatured at 94˚C for 30 s, denatured at 94˚C for 30 s, annealed at 68˚C for 30 s, extended at 72˚C for 2 min, 12 cycles, fully extended at 72˚C for 10 min.

After two rounds of hybridization and nested PCR amplification, the purified forward and reverse PCR products were linked to pUCm-T clone vector and then transformed into DH5α competent cells. After transformation, positive transformants were screened by ampicillin antibiotics and blue-white spot assay. 400 white clones were randomly selected from the forward and reverse libraries, the size of the insert fragment and the uniqueness of the band were detected by bacteria liquid PCR. The reaction system of bacteria liquid PCR was as follows: 1μL bacterial fluid, 0.5μL Nested PCR primer 1, 0.5μL Nested PCR primer 2R, 5μL 2×TsingKe master mix, 3μL deionized water. The mixtures were pre-denatured at 95˚C for 1 min, denatured at 95˚C for 30 s, annealed at 60˚C for 30 s, extended at 72˚C for 30 s, 30 cycles, fully extended at 72˚C for 10 min.

## Sequencing and sequence analysis

After the positive clones were amplified in liquid medium, the sequencing reactions were entrusted to Beijing TsingKe Biological Technology Co., Ltd. The sequencing results were compared by the BLAST function of the National Center for Biotechnology Information (NCBI; http://www.ncbi.nlm.nih.gov). We counted valid ESTs and got forward and reverse SSH gene libraries after removing redundant sequences, vector sequences and sequences that were unmatched. The SSH libraries were analyzed by GO annotation through Blast2Go software version 5.2, and the immune-related genes were screened.

## Verification of differentially expressed genes by qRT-PCR

The immune-related differential genes of the SSH libraries were selected, and we designed a fluorescent quantitative primer for them by the online primer design tool Primer Quest Tool (https://sg.idtdna.com). Plasmid standards of differentially expressed genes were prepared using plasmid pUCm-T. Then the copies of the plasmid standards were calculated, and they were diluted 10-fold to 7 gradients with deionized water. Bio-Rad CFX 96 was used to make the standard curve. At the same time, the differential genes were detected by absolute qPCR using single-stranded cDNA diluted 10-fold as template. *β-actin* was used as a reference gene to consider the variation of total input cDNA. The reaction system was as follows: 1μL diluted single-stranded cDNA, 10μL SYBR Green supermix, 1μL forward primer, 1μL reverse primer, supplemented with RNase-free water to 20μL. Each reaction was repeated three times. The mixtures were pre-denatured at 94˚C for 3 min, denatured at 94˚C for 5 s, annealed at suitable annealing temperature for 30 s, extended at 72˚C for 15 s, 40 cycles. Bio-Rad CFX Manager 3.1 software was used to analyze the qPCR results, the number of copies of differential

**Table 2. qPCR primer design of immune-related differentially expressed genes.**

| Gene | Forward primer | Reverse primer | Annealing temperature (°C) | Product length (bp) |
|------|---------------|----------------|---------------------------|---------------------|
| *JCHAIN* | 5'-GGTTCGTCCTTGT GGCAGGTTATC-3' | 5'-GAGGTCACCGTTA CGCACTTACAC-3' | 58 | 88 |
| *FTH1* | 5'-ATGTGACCAACCT GCGGAAGATG-3' | 5'-TGCATTGCTGGACC AGTGAAGTAG-3' | 60 | 156 |
| *P2RX7* | 5'-AGTTCGCGTTAC CCTGAAAG-3' | 5'-TCTCTTGTCTGCGT TGGTATG-3' | 59 | 84 |
| *SLC7A6* | 5'-GTCTTTGGAGCT CTCTGCTATG-3' | 5'-AGTGAGGTCCACA AACGAATAA-3' | 58 | 122 |
| *TLR7* | 5'-ACCGTCGCCTCAA GGAAGTCC-3' | 5'-ACGCAGTTGCACC TGAAGTCAATC-3' | 57 | 145 |
| *IGF1R* | 5'-GGACACAGAGGA GCTTGACC-3' | 5'-TGTCAGTGGGTTG GAGGGTA-3' | 58 | 83 |
| *SMAD7* | 5'-CTGGTGCGTGGTG GCATACTG-3' | 5'-CTGGCTTCTGTTGT CCGAGTTGAG-3' | 58 | 145 |
| *ITGA4* | 5'-ACAGAAGAAGGC AGGTGAATAG-3' | 5'-GGCAGAACACAGA GTATGTAGG-3' | 59 | 124 |
| *DOCK10* | 5'-GGCCACAGCTCAG ATGAAGGAAC-3' | 5'-TGAGAGCAGCGAT GTGAATGTAGC-3' | 60 | 191 |
| *β-actin* | 5'-TGCTGTGTTCCCA TCTATCG-3' | 5'-TTGGTGACAATAC CGTGTTCA-3' | 58 | 150 |

genes was calculated by standard curve. The qPCR primer sequences and optimal annealing temperature of the differential genes are shown in Table 2.

## Statistical analysis

All the experimental data were analyzed by one-way ANOVA with SPSS 23.0 software and Duncan's method was used for multiple comparisons, with $P < 0.05$ as the statistically significant, $P < 0.01$ as the highly statistically significant.

# Results

## Construction of cDNA libraries

The results of 1.2% agarose gel electrophoresis of double-stranded cDNA products under different cycles is shown in Fig 1A and 1B (S1 Fig, S1 Raw images). Treated group have the brightest bands and the widest range in 24 cycles. Control group have the brightest bands in 27 cycles, but the widest range in 24 cycles. As can be seen from the figure, the optimal cycles for both treated group and control group was 24.

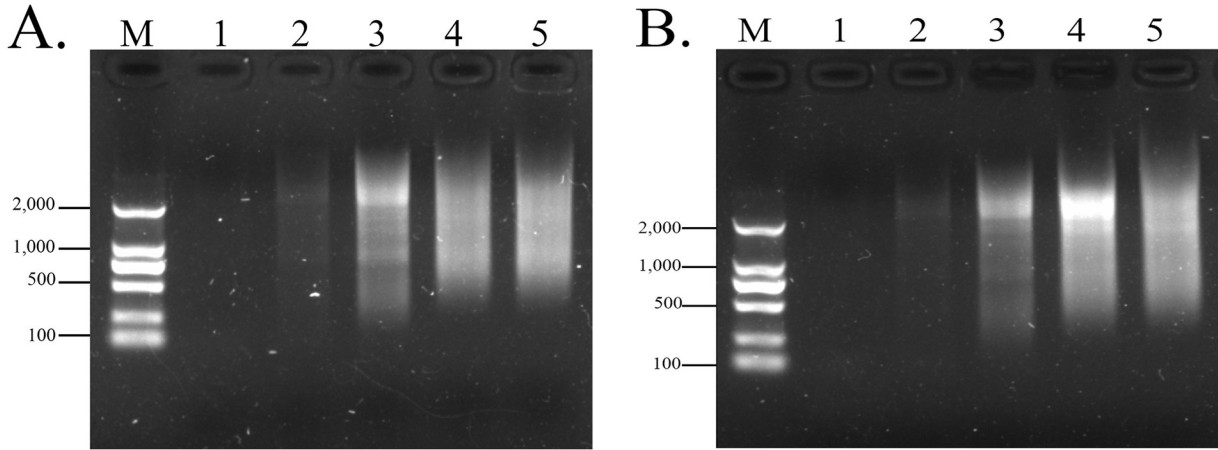

**Fig 1. Agarose electrophoresis analysis of optimal cycles of double-stranded cDNA.** Electrophoresis with agarose of 1.2% concentration. The images were generated by the Gel imaging system Gel Doc™ XR+. (A) Treated group. (B) Control group. M represents DNA Marker 2000. The numbers 1, 2, 3, 4 and 5 represent 18, 21, 24, 27 and 30 PCR cycles respectively.

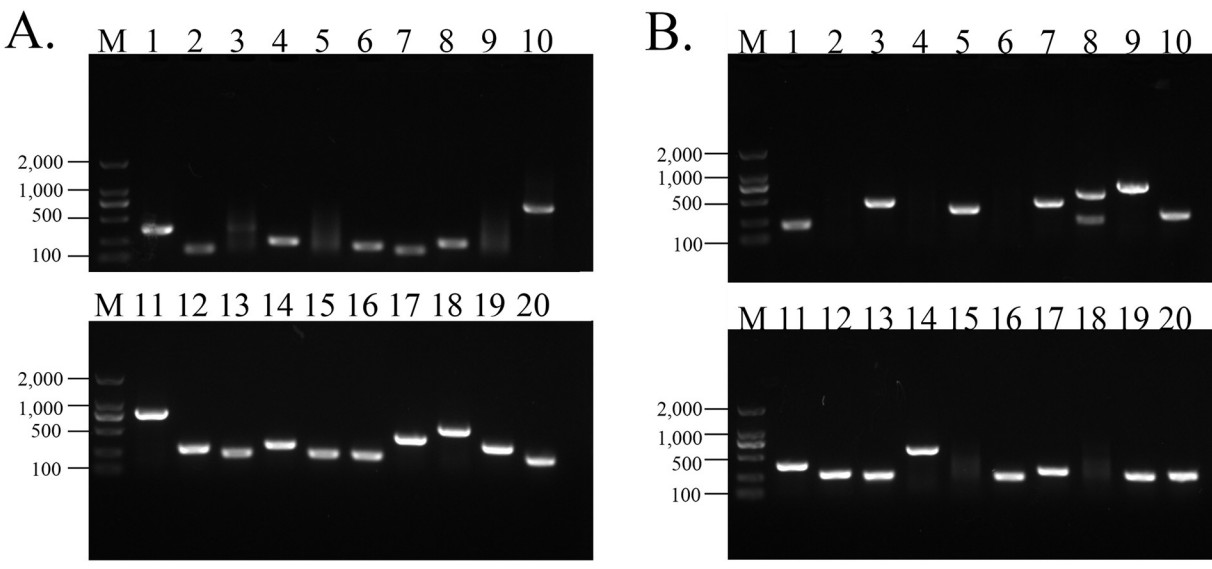

**Fig 2. Detection of inserted fragments by bacteria liquid PCR.** Electrophoresis with agarose of 1.2% concentration. The images were generated by the Gel imaging system Gel Doc™ XR+. (A) Treated group. (B) Control group. M represents DNA Marker 2000. The numbers 1, 2, 3, et al. represent different bacterial clones.

## Construction of SSH libraries

200 white clones were randomly selected from the forward and reverse libraries respectively. Partial result of bacteria liquid PCR is presented in Fig 2A and 2B (S2 Fig, S2 Raw images), most of them are positive clones, and the size of the bands is concentrated at 200 bp to 1000 bp, which conforms to the enzyme digestion effect and meets the requirements of the suppression subtractive hybridization libraries.

## Sequencing and sequence analysis

In the forward and reverse libraries, 129 and 155 positive clones were successfully sequenced, respectively. Excluding the low-quality sequences, the unmatched sequences, the repeat sequences, and the unannotated protein sequences, 63 and 56 valid ESTs were obtained, respectively (S2 Table), respectively. These EST sequences were submitted to NCBI's dbEST database with the GenBank accession numbers JZ980559-JZ980677.

After clustering analysis of Gene ontology on Level 2, data showed that in the forward subtracted cDNA library, within the biological process (GO:0008150) category, 63 ESTs were classified into 14 categories, comprising: cellular process (15%), metabolic process (12%), biological regulation (10%), regulation of biological process (10%), response to stimulus (9%), signaling (8%), and multicellular organismal process (6%), localization (6%), cellular component organization or biogenesis (6%), developmental process (6%), etc. (Fig 3A). Within the cellular component (GO:0005575) category, they were classified into 8 categories, comprising: cells (23%), cell part (22%), organelle (13%), membrane (12%), organelle part (9%), membrane part (9%), protein-containing complex (9%) etc. (Fig 3B). Within the molecular function (GO:0003674) category, they were classified into 5 categories, comprising: binding (54%), catalytic activity (22%), molecular transducer activity (10%), transcription regulator activity (7%) and transporter activity (6%) (Fig 3C). In the reverse subtracted cDNA library, within the biological process category, 56 ESTs were classified into 20 categories, comprising: cellular process (15%), metabolic process (13%), biological regulation (10%), regulation of biological process

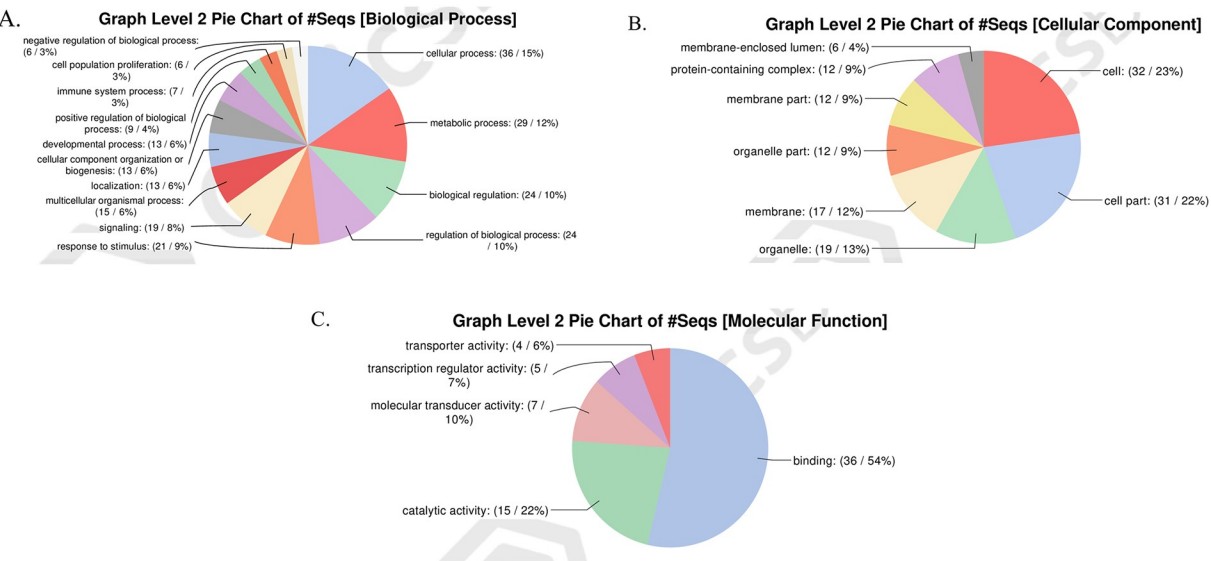

**Fig 3. Functional classification of genes in forward library.** (A) Classification of biological processes level. (B) Classification of cell component level. (C) Classification of molecular function level.

(10%), response to stimulus (7%), developmental process (6%), positive regulation of biological process (5%) and localization (5%) etc. (Fig 4A). Regarding the cellular component category, they were classified into 10 categories, including cells (21%), cell part (20%), organelle (18%), protein-containing complex (11%), organelle part (10%), membrane-enclosed lumen (9%) and membrane (5%) etc. (Fig 4B). In the molecular function category, they were classified into 3 categories, including binding (63%), catalytic activity (29%) and transcription regulator activity (8%) (Fig 4C).

Through GO cluster analysis, immune-related genes were screened including joining chain of multimeric IgA and IgM (*JCHAIN*), ferritin heavy chain 1 (*FTH1*), purinergic receptor P2X7 (*P2RX7*), solute carrier family 7 member 6 (*SLC7A6*), toll like receptor 7 (*TLR7*), insulin like growth factor 1 receptor (*IGF1R*), SMAD family member 7 (*SMAD7*), integrin subunit alpha 4 (*ITGA4*) and dedicator of cytokinesis 10 (*DOCK10*) (Table 3).

## Verification of differentially expressed genes by qRT-PCR

Absolute qPCR was used to detect the expression levels of nine immune-related differentially expressed genes in spleen tissues. According to the standard curve of qPCR of different genes, the copies of the genes were calculated (S3–S12 Figs). The results showed that the expression level of the reference gene *β-actin* was almost the same (*P>0.05*) in the treated group and control group, and the copies of seven up-regulated genes in the treated group were obviously higher than that in the control group, the copies of the two down-regulated genes in the treated group were less than that in the control group. Among them, the expression levels of *JCHAIN*, *FTH1*, *P2RX7*, *TLR7*, *IGF1R*, *SMAD7* in the treated group were highly significantly (*P<0.01*) higher than those in the control group, and *SLC7A6* was significantly (*P<0.05*) higher than that in the control group. Meanwhile, the expression levels of *DOCK10* in the treated group was highly significantly (*P<0.01*) lesser than that in the control group, and *ITGA4* was significantly (*P<0.05*) lesser than that in the control group (Fig 5, S13 Fig). These results were consistent with those of SSH.

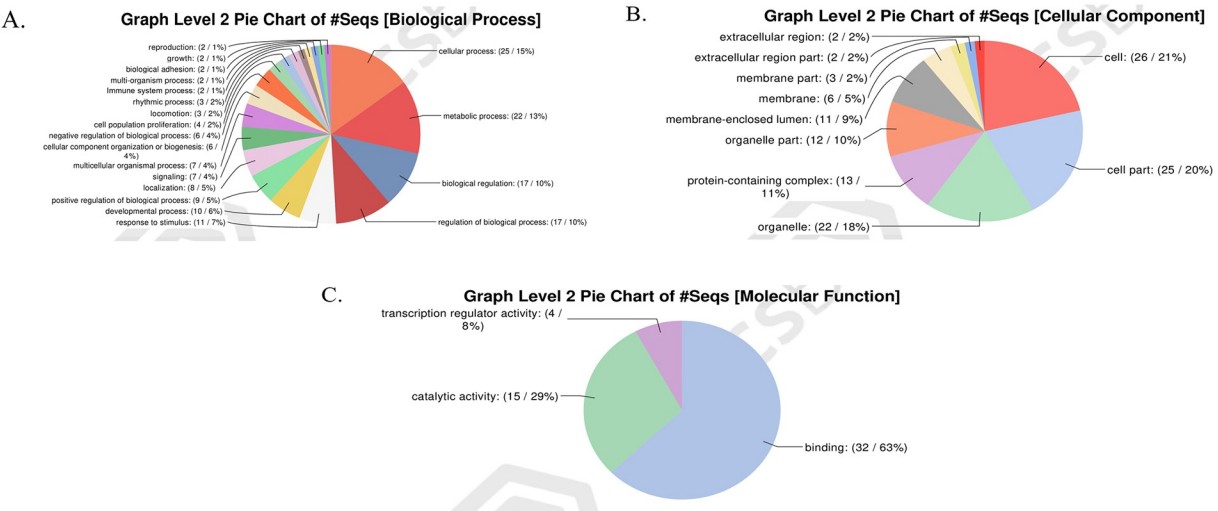

**Fig 4. Functional classification of genes in reverse library.** (A) Classification of biological processes level. (B) Classification of cell component level. (C) Classification of molecular function level.

## Discussion

In the field of veterinary medicine, SSH has been used to screen the differentially expressed genes related to animal immunity in recent years. Gao et al. [15] studied the immune mechanism of silkworm against Bombyxmori nuclear polyhedrosis virus (*BmNPV*), the differentially expressed gene libraries of resistant strains and susceptible strains were constructed by SSH technique, and they found that 17 genes were up-regulated in resistant strain BC10, which involved in cell metabolism, transmembrane transport, cytoskeleton, protease, development and immunity, and these genes may be related to resistance to *BmNPV* in silkworm. In order to study the effect of highly pathogenic porcine reproductive and respiratory syndrome virus (*HP-PRRSV*) infection on the response of pig cells, SSH was used to compare cDNA libraries of porcine alveolar macrophages (PAM) infected with *HP-PRRSV* and uninfected *HP-PRRSV*, and results showed that 21 genes including *IL-16*, TGF-β type 1 receptor, epidermal growth factor, MHC-I SLA, toll-like receptor, hepatoma-derived growth factor, *FTH1*, and MHC-II SLA-DRB1 were up-regulated in PAM infected with *HP-PRRSV*, by the way, this research demonstrated differential gene expression between *HP-PRRSV*-infected and uninfected PAMs *in vivo* for the first time [16].

**Table 3. Differentially expressed genes related to immunity screened by SSH.** (Treated group vs Control group).

| Gene | Accession number | Up / Down regulation |
| --- | --- | --- |
| Joining chain of multimeric IgA and IgM (*JCHAIN*) | NM_204263.1 | up |
| Ferritin heavy chain 1 (*FTH1*) | NM_205086.1 | up |
| Purinergic receptor P2X 7 (*P2RX7*) | XM_001235162.5 | up |
| Solute carrier family 7 member 6 (*SLC7A6*) | XM_025154295.1 | up |
| Toll like receptor 7 (*TLR7*) | NM_001011688.2 | up |
| Insulin like growth factor 1 receptor (*IGF1R*) | NM_205032.1 | up |
| SMAD family member 7 (*SMAD7*) | XM_004949014.3 | up |
| Integrin subunit alpha 4 (*ITGA4*) | XM_421974.6 | down |
| Dedicator of cytokinesis 10 (*DOCK10*) | XM_015277026.2 | down |

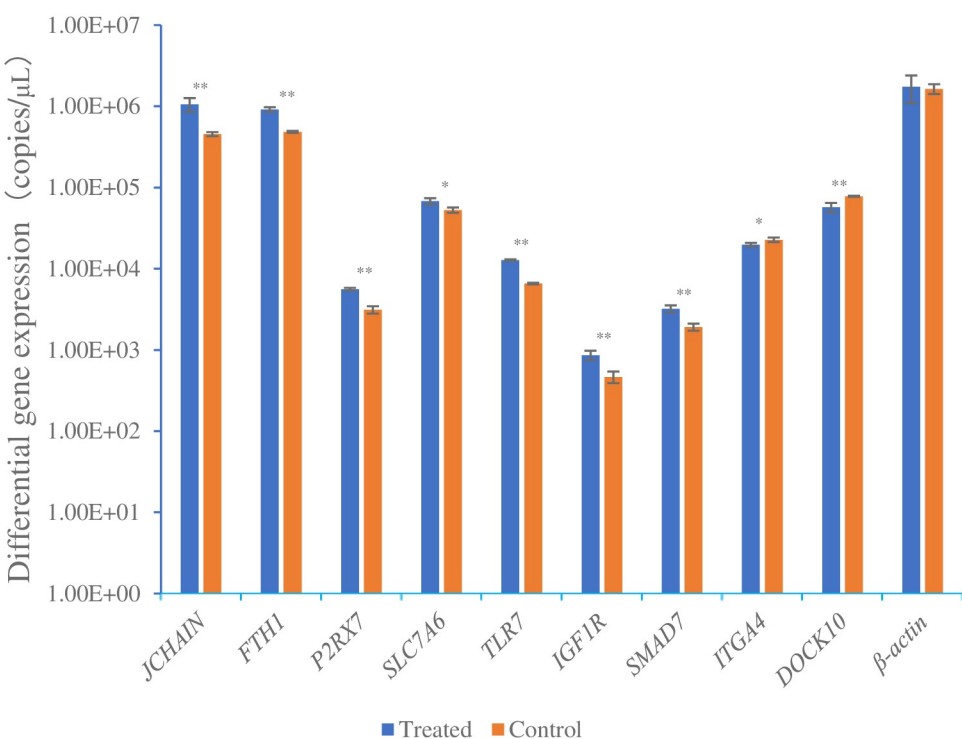

**Fig 5. Detection of differentially expressed genes by absolute qPCR.** Experimental data were analyzed by one-way ANOVA with SPSS 23.0 software and Duncan's method was used for multiple comparisons * *P<0.05*, ** *P<0.01*.

In present study, 284 positive clones of SSH libraries were sequenced successfully and 119 valid ESTs were obtained by eliminating duplication. Among them, 63 ESTs belong to forward library and 56 ESTs belong to reverse library. Through GO cluster analysis and qPCR identification, 9 immune-related differentially expressed genes were screened out, including *JCHAIN*, *FTH1*, *P2RX7*, *SLC7A6*, *TLR7*, *IGF1R*, *SMAD7*, *ITGA4*, *DOCK10*. Seven of the nine genes were up-regulated and two genes were down-regulated in the broilers fed probiotic *Bacillus cereus* PAS38.

*JCHAIN* is a glycoprotein about 15 kDa in size, it can initiate the polymerization process by linking with IgM or IgA caudal cysteine and covalently bind to poly IgM or IgA stably by disulfide bond. It plays an important role in the secretion, transportation of immunoglobulin and the activation of complement [17, 18]. *JCHAIN* has highly conserved structural characteristics in various vertebrates and invertebrates. Study has found that the amino acid composition of *JCHAIN* in chickens and humans is highly similar [19]. Takahashi et al. used Northern blot hybridization to study the expression level of *JCHAIN* in brain, intestine, thymus, spleen, bursa of fabricius and other tissues and organs of adult chickens. It was found that the expression level of *JCHAIN* in spleen and rectum was relatively high, but in thymus was relatively low [20]. Previous studies have shown that after lactating women ingested daily viable *Lactobacillus bulgaricus* for 28 days [21], probiotics activate the lymphocytes and synthesize IgA dimers with J-chain, which increased SIgA content in both breast milk and feces obviously. In present study, *JCHAIN* expression in the spleen of broilers was highly significantly increased after feeding *Bacillus cereus* PAS38, which may play an important role in the expression of downstream cytokines, thus affecting the immunity of broilers.

The expression of *FTH1* in the spleen of broilers was also highly significantly increased after feeding *Bacillus cereus* PAS38 in this experiment. *FTH1* is approximately 21kDa in size and is the major functional subunit of ferritin. It contains a $Fe^{2+}$ oxidation center responsible for the oxidation and integration of iron ions to maintain balance of iron ions in the body [22, 23]. Previous studies had found that *FTH1* was expressed differently in different breeds of chickens. Willson et al. found that the expression of *FTH1* in laying hen strain was higher than that in broiler strain when they studied the difference in liver transcriptome between broilers and layers [24]. At the same time, Yang et al. studied the difference in ovarian transcriptome between low-yielding laying hens and high-yielding laying hens, and found that the expression of *FTH1* in high-yielding laying hens was lower than that in low-yielding laying hens [25]. Which suggested that proper expression of *FTH1* may play an important role in laying mechanism of chickens. *FTH1* is not only related to laying traits of chickens, but also to immune defense. Matulova et al. infected chickens with *Salmonella enteritidis*, found that *FTH1* in the spleen of chickens was significantly up-regulated [26]. On the contrary, Niroshan et al. infected chickens with Marek's disease virus, found that *FTH1* protein in the spleen of chickens was down-regulated [27]. These distinct results suggest that chickens may have different defense mechanisms against bacterial and viral infections. Zhang et al. [28] found that the expression of *FTH1* in liver and spleen of Jingding duck was significantly down-regulated after infection with duck hepatitis virus 1, and the expression of antiviral marker gene *MX1* was significantly up-regulated in DF-1 cells transfected with the *FTH1* plasmid. Sun et al. [29] showed that after *FTH1* silencing, the intracellular viability of *Brucella* decreased, and the apoptotic rate of macrophages increased. These studies suggest that *FTH1* can play a role in resisting inflammation in the process of virus and bacteria invading the organism.

Purinergic receptor P2X7 is an important member of purine receptor family. *P2RX7* is mainly involved in cell signal transduction, cytokine secretion, cell proliferation and apoptosis [30]. *P2RX7* plays an important role in bone metabolism pathway [31]. Zhang et al. added 10 mg/kg icariin to drinking water of broilers with tibial cartilage dysplasia, and found that the expression of *P2RX7* increased significantly, and the mortality and lameness rate of broilers were also reduced [32]. Meanwhile, it is also involved in the immune response by mediating the NF-κB pathway and the NALP3 inflammatory pathway [33]. Ren et al. [34] conducted transcriptome analysis on the liver of chickens infected with avian adenovirus type 4, and the results showed that *P2RX7* was up-regulated after 21 days of infection. Similarly, Kalenik et al. [35] showed that *P2RX7* in spleen was significantly up-regulated after inoculation of chickens with three H5N1 subunit vaccines. According to the results of absolute qPCR, the *P2RX7* was highly significantly up-regulated after feeding with *Bacillus cereus* PAS38, the gene may play an important role in the spleen immune response of broilers.

*SLC7A6* is one of 14 members of solute carrier family 7, it generally works with other members of the solute carrier family to transport amino acids and glycoproteins in many cells. The current research on *SLC7A6* focuses on its potential applications in various types of disease in humans [36, 37]. The difference in expression of *SLC7A6* in male and female broilers has also been studied by Kaminski et al. [38], the results showed that the expression of *SLC7A6* in the intestine of male chickens was significantly higher than that of female chickens. However, the role of this gene in immune response is still unclear. In this experiment, the *SLC7A6* of broilers fed with *Bacillus cereus* PAS38 was significantly up-regulated. Considering the function of amino acid transport of the gene, we speculated that *SLC7A6* might improve the immune ability of broilers by promoting the development of immune organs.

*TLR7* is a common toll-like receptor expressed in endosome. It can induce the secretion of cytokines such as INF-γ by mediating the signal pathway of NF-κB and play an important role in natural immunity [39]. Gupta et al. [40] found that *TLR7*-based adjuvants combined with

the influenza A vaccine can stimulate the expression of *IFN-α*, *IFN-β* and other genes in chicken spleen cells and inhibit the replication of influenza A. And *TLR7*-based adjuvants combined with the *S. Enteritidis* antigen can reduced liver and spleen organ invasion by S. Enteritidis in chickens. Xiang et al. [41] infected mature chicken bone-marrow-derived dendritic cells with GM strain of Newcastle disease virus (NDV) and found that *TLR7* was significantly up-regulated. Regulation of intestinal epithelial immunological function by the probiotic *Lactobacillus johnsonii* N6.2 was shown using human Caco-2 cell monolayers, *TLR7* expression levels were upregulated by *L. johnsonii* N6.2 [42]. Guo et al. [43] found that the expression of *TLR7* in the spleen of cherry valley duck was up-regulated by adding *Bacillus subtilis* into the diet. These studies are similar with the results of this experiment, indicating that *Bacillus cereus* PAS38 can stimulate the immune cells and make them in a high state of immune alertness.

As one of the receptors of IGF1, *IGF1R* has a strong role in promoting the growth and differentiation. Among chickens and other birds, *IGF1R* is a unique receptor of IGF-I and IGF-II. It plays a very important role in the function of IGFs and is an important candidate gene affecting chicken growth and body composition [44]. Alzaid et al. [45] found that the expression of *IGF1R* in head kidney and spleen of salmon was significantly down-regulated after salmonella infection, and *IGF* was also down-regulated. It was suggesting that *IGF1R* may have an important connection with immune pathways of animals. Giorgia et al. [46] found that tilapia was fed with *Lactobacillus rhamnosus* preparation, it could significantly increase the expression of *IGF1R* in muscles, which was similar with this experiment. At present, the research of *IGF1R* in chickens is mainly about the effect of its polymorphism on the growth and breeding of chickens. Wang et al. [47] found that selenium deficiency can decreases the growth rate of spleen and the number of splenic lymphocytes by deactivating the IGF-1R/PI3K/Akt/mTOR pathway in chickens. These studies indicate that *IGF1R* is also indispensable in chicken immune system.

*SMAD7* is an endogenous negative feedback regulator of transforming growth factor β / bone morphogenetic protein (TGF-β / BMP) signaling pathway [48]. Studies have shown that high expression of *SMAD7* can resist fibrosis and inflammation mediated by *TGF-β* signaling pathway [49], and the absence of *SMAD7* may affect the development of splenic dendritic cells (DCs) [50]. It indicates that the expression of *SMAD7* affects the immune function of the body. It was found that *SMAD7* plays an important role in the differentiation of embryonic stem cells and muscle development in chicken [51,52], but there are few reports about its role in chicken immunity, which needs further study. Tohru et al. [53] found that after preterm infants were treated with *Bifidobacterium breve*, the expression of *SMAD3* in serum increased, it is similar with this experiment, which suggested that *Bacillus cereus* PAS38 may play an active role in the immune process of broilers.

*ITGA4* is one of α subunits of integrin. Integrins are a class of glycoproteins, which consist of a α subunit and a β subunit. They are important cell surface receptors. They mediate adhesion of cells to extracellular matrices and adhesion of cell to cell, participate in cell proliferation, differentiation, adhesion, migration and other processes, and play an important role in the growth and development, immune response and other physiological processes of the body [54, 55]. Kim et al. [56] found that the down-regulation of *ITGA4* in duodenum of chickens infected with *Eimeria*. On the contrary, Heidar et al. [57] infected susceptible chickens and resistant chickens respectively with Marek's disease virus (MDV), and found that *ITGA4* was up-regulated to a certain extent in duodenum, and especially in resistant chickens. As a cell adhesion related molecule, *ITGA4* is involved in leukocyte migration, T cell adhesion and T cell death. These two different results may be related to the tolerance of chickens to the antigens. In this experiment, after feeding *Bacillus cereus* PAS38, the *ITGA4* in spleen of broilers

was down regulated, whether this is a positive signal of immunity is unknown, and its specific mechanism needs further study.

In this experiment, compared with the control group, the expression of *DOCK10* was also down regulated. *DOCK10* is a member of the dock family and can activate Rac1, a small GTP protease [58]. It has the function of regulating the activation and migration of immune cells [59]. Studies have found that the absence of *DOCK10* could reduce the number of B lymphocytes in the spleen and peripheral blood, suggesting that *DOCK10* may be associated with the maturation and activation of B cells [60, 61]. Monson et al. [62] studied the effect of aflatoxin B1 and probiotics on turkey spleen transcriptome, and found that the expression level of *DOCK10* had no significant different in the group of aflatoxin B1 and the group of probiotics compared with the control group, but it was significantly up-regulated in the group of aflatoxin B1+probiotics compared with the control group. Ansari et al. [63] investigated the mechanisms that induce atrophy of the chicken bursa of fabricius upon lipopolysaccharide (LPS) treatment in young chicks, found that LPS treatment resulted in a significant decrease in the weight of bursa of fabricius, and the decrease of cell proliferation; at the same time, the *DOCK10* was significantly up-regulated. This suggests that *DOCK10*, as a proinflammatory factor, its high expression can induce abnormal apoptosis. In this experiment, the decrease of *DOCK10* may be related to the dosage of *Bacillus cereus* PAS38, on the other hand, probiotics do inhibit individual innate immune genes, but we need to know that the beneficial effect of probiotics on animal immune system has been widely proved. And next, we will continue to pay attention to and study the mechanism of *DOCK10* in broiler immunity.

The effect of probiotics on the expression of immune-related genes may be related to the process of immune system stimulation by its surface antigens [64, 65]. However, the exact signaling pathway remains unknown, and there are few reports on how these nine differential genes affect the immunity of broilers. In the future, methods such as gene silencing or knock-out and gene overexpression will be needed to further explore the effects of these genes on function of immune organ in broilers.

## Conclusion

The differential expression of immune-related genes in spleen of broilers fed with probiotic *Bacillus cereus* PAS38 preparation was studied by SSH technique, nine differentially expressed immune-related genes were screened out. Absolute qRT-PCR was used to verify the immune-related differentially expressed genes, it was found that *JCHAIN*, *FTH1*, *P2RX7*, *TLR7*, *IGF1R*, *SMAD7*, and *SLC7A6* were significantly up-regulated in the treated group. These results suggest that *Bacillus cereus* PAS38 preparation may improve the immunity of broilers by regulating the expression of some immune-related genes, and provide useful information for further understanding the molecular mechanism of probiotics affecting poultry immunity.

## Supporting information

**S1 Fig. Agarose electrophoresis analysis of optimal cycles of double-stranded cDNA.** Electrophoresis with agarose of 1.2% concentration. The images were generated by the Gel imaging system Gel Doc™ XR+. (A) Treated group. (B) Control group. M represents DNA Marker 2000. The numbers 1, 2, 3, 4 and 5 represent 18, 21, 24, 27 and 30 PCR cycles respectively. Fig 1A was generated by S1A Fig, and Fig 1B was generated by S1B Fig.
(TIF)

**S2 Fig. Detection of inserted fragments by bacteria liquid PCR.** Electrophoresis with agarose of 1.2% concentration. The images were generated by the Gel imaging system Gel Doc™ XR+.

(A) Treated group. (B) Control group. M represents DNA Marker 2000. The numbers 1, 2, 3, et al. represent different bacterial clones. Fig 2A was generated by S2A Fig, and Fig 2B was generated by S2B Fig.
(TIF)

**S3 Fig. Standard curve of *JCHAIN*.** The abscissa represents the concentration of plasmid standard (Log$_{10}$N copies/μL). The longitudinal coordinates denote the cycle threshold. The same below.
(TIF)

**S4 Fig. Standard curve of *FTH1*.**
(TIF)

**S5 Fig. Standard curve of *P2RX7*.**
(TIF)

**S6 Fig. Standard curve of *SLC7A6*.**
(TIF)

**S7 Fig. Standard curve of *TLR7*.**
(TIF)

**S8 Fig. Standard curve of *IGF1R*.**
(TIF)

**S9 Fig. Standard curve of *SMAD7*.**
(TIF)

**S10 Fig. Standard curve of *ITGA4*.**
(TIF)

**S11 Fig. Standard curve of *DOCK10*.**
(TIF)

**S12 Fig. Standard curve of *β-actin*.**
(TIF)

**S13 Fig. The process of calculating the copies of differentially expressed genes in absolute qRT-PCR.** P-value is calculated by SPSS 23.0 software.
(TIF)

**S1 Table. Composition of solid fermentation medium of *Bacillus cereus* PAS38.**
(PDF)

**S2 Table. Alignment results of sequencing clones on NCBI.**
(PDF)

**S1 Raw images.**
(PDF)

**S2 Raw images.**
(PDF)

## Acknowledgments

We thank the Microecology Research Laboratory for providing experimental instruments, and Abdul Khalique for the support to the manuscript revision.

## Author Contributions

**Conceptualization:** Jiajun Li, Wanqiang Li, Jianzhen Li, Kangcheng Pan.

**Data curation:** Jiajun Li, Wanqiang Li, Jianzhen Li, Zhenhua Wang, Dongmei Zhang.

**Formal analysis:** Jiajun Li, Wanqiang Li, Jianzhen Li, Zhenhua Wang, Dan Xiao, Yufei Wang, Xueqin Ni, Dong Zeng.

**Funding acquisition:** Dongmei Zhang.

**Investigation:** Jiajun Li, Wanqiang Li, Dan Xiao, Xueqin Ni.

**Methodology:** Jiajun Li, Wanqiang Li, Jianzhen Li, Zhenhua Wang, Dan Xiao, Yufei Wang, Xueqin Ni, Dong Zeng, Dongmei Zhang, Bo Jing, Lei Liu, Qihui Luo, Kangcheng Pan.

**Project administration:** Jiajun Li, Jianzhen Li, Zhenhua Wang, Dan Xiao, Yufei Wang.

**Resources:** Jianzhen Li, Bo Jing.

**Software:** Jiajun Li.

**Supervision:** Wanqiang Li, Dan Xiao, Yufei Wang, Xueqin Ni, Dong Zeng, Dongmei Zhang, Bo Jing, Lei Liu, Qihui Luo.

**Validation:** Jiajun Li, Yufei Wang.

**Writing – original draft:** Jiajun Li.

**Writing – review & editing:** Jiajun Li, Wanqiang Li, Jianzhen Li, Zhenhua Wang, Dan Xiao, Yufei Wang, Xueqin Ni, Dong Zeng, Dongmei Zhang, Bo Jing, Lei Liu, Qihui Luo, Kangcheng Pan.

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
