## [Decision Letter · Decision Letter 0]

1 Oct 2019

PONE-D-19-18103

Screening of differentially expressed immune-related genes from spleen of broilers fed with probiotic Bacillus cereus PAS38 based on suppression subtractive hybridization

PLOS ONE

Dear Dr. PAN,

Thank you for submitting your manuscript to PLOS ONE. After careful consideration, we feel that it has merit but does not fully meet PLOS ONE’s publication criteria as it currently stands. Therefore, we invite you to submit a revised version of the manuscript that addresses the points raised during the review process.

We would appreciate receiving your revised manuscript by Nov 15 2019 11:59PM. To enhance the reproducibility of your results, we recommend that if applicable you deposit your laboratory protocols in protocols.io, where a protocol can be assigned its own identifier (DOI) such that it can be cited independently in the future. For instructions see: http://journals.plos.org/plosone/s/submission-guidelines#loc-laboratory-protocols

We look forward to receiving your revised manuscript.

Kind regards,

Juan J Loor

Academic Editor

PLOS ONE

Journal Requirements:

2. To comply with PLOS ONE submissions requirements, please provide methods of sacrifice in the Methods section of your manuscript.

Additional Editor Comments (if provided):

Reviewers' comments:

Reviewer's Responses to Questions

**Comments to the Author**

1. Is the manuscript technically sound, and do the data support the conclusions?

Reviewer #1: Yes

Reviewer #2: Partly

2. Has the statistical analysis been performed appropriately and rigorously? 

Reviewer #1: Yes

Reviewer #2: Yes

3. Have the authors made all data underlying the findings in their manuscript fully available?

Reviewer #1: Yes

Reviewer #2: No

4. Is the manuscript presented in an intelligible fashion and written in standard English?

Reviewer #1: Yes

Reviewer #2: Yes

5. Review Comments to the Author

Reviewer #1: The Authors aimed to construct the spleen differential genes library of broilers fed with probiotic Bacillus cereus PAS38 by suppression subtractive hybridization and screen the immune-related genes. The Authors have investigated an interesting topic and the theme has been properly described.

I would like to congratulate authors for the good-quality of the article, the literature reported used to write the paper, and for the clear and elegant and appropriate structure. The manuscript is well written, presented and discussed, and understandable to a specialist readership.

In general, the organization and the structure of the article are satisfactory and in agreement with the journal instructions for authors. The subject is adequate with the journal scope.

The work shows a conscientious study in which a very exhaustive discussion of the literature available has been carried out. The introduction provides sufficient background, and the other sections include results clearly presented and analyzed exhaustively.

As specific comment, I suggest to supply more high-quality figures.

Reviewer #2: This manuscript used suppression subtractive hybridization to identify genes that could have differential expression in response to the oral probiotic Bacillus cereus PAS38. They identified 119 valid ESTs and focused on 9 with immune related functions. Of these, only 3 were validated by qRT-PCR to have a treatment effect.

1. Why was SSH selected as the method for predicting differentially expressed genes? Since the chicken genome is fairly well annotated, why was RNA-seq or 3’ Quant-Seq (better suited to large sample numbers) not used? These methods can be statistically tested (unlike SSH) and also screen the whole transcriptome without a priori knowledge of the treatment effect.

2. Perhaps the low confirmation rate of the qRT-PCR reflects the lack of normalization to a reference gene. Even for absolute qRT-PCR, the copy number for each test gene needs to be normalized to a reference gene to account for variation in total input RNA. Please add the reference information if already done, or complete an additional qRT-PCR using a stably expressed reference gene, adjust the copy number for the 9 target genes and run the ANOVA again.

3. Please strengthen the evidence in the introduction for Bacillus cereus as a probiotic.

4. The discussion included very few references for chicken, with none for the significant JCHAIN, FTH1, and P2RX7 genes. Please add literature for these 3 genes in chicken.

5. The GO term results in lines 149-152 are not interpreted clearly. As analyzed, the biological process, molecular function and cell component terms are considered as if they are mutually exclusive and each EST can only have 1 type of term. It would be better to report this information as the percentage of the total ESTs in each category (so each EST can be represented 0-3 times) rather than a percentage of total GO terms.

6. Figure S6 and S8 show that IGF1R and ITGA4 primers have really low efficiency (60%). Were any other primer pairs tested for these genes? Perhaps an alternate would amplify better and better discriminate between the treated and control groups.

7. A gene set enrichment analysis (such as with PANTHER or GSEA) might be more informative than the top level GO terms from DAVID.

8. Based on the feeding protocol, there were 30 control and 30 treated birds. How many spleens from each group were used to perform the SSH and were the samples pooled? How many individual samples/group were used in the qRT-PCR validation? Add to the text and to figure legends. Also note that if all 60 birds were used, the phrase “randomly selected” in line 81 doesn’t make sense.

9. There are many other experimental details missing from the methods section, including:

a. What is the “basic diet”? Please move Table S3 into the main manuscript, as the composition of the diet is important to the interpretation of this study.

b. Please put the information in Table S1 and S2 into the text of the methods and not supplemental tables, so that the company and reagent information is complete in the text.

c. What type of housing (floor pens, cages, etc.) was used for the birds?

d. Please add the bacterial growth conditions (media, temp, etc.), and how the bacteria concentration was determined to line 72-74.

e. In line 87, “diluted 5 times” is not clear. Please clarify if this means a 5-fold (1:5) dilution or 5 serial dilutions to what final dilution factor.

f. How much input RNA was used for cDNA synthesis in line 87 and 93-94? How much cDNA was used as input for the qRT-PCR reactions in line 116?

g. In line 98, what were the hybridization conditions? Also add to this section, the enzyme and condition used for digestion, as mentioned in line 138.

h. For all reactions (cDNA synthesis in line 87 and 93-94, nested PCR in line 98, liquid-culture treated PCR in line 102-103, and qRT-PCR in line 116), the components of the reactions and the cycling conditions (i.e. the program) need to be described. For the qRT-PCR, please also include a description of how the standard curves were generated/used and reference the supplemental figures.

i. Were amplicons from the qRT-PCR confirmed by sequencing?

j. In line 110, what version of the DAVID database was used?

k. What were the positive and negative controls for the SSH and qRT-PCR validation?

l. The percentage agarose used for electrophoresis is not consistent. Is it 1.2% (line 90 and 128) or 1.5% (line 133 and 141)?

10. Other comments:

a. There are English grammar errors in the manuscript, please check.

b. In Figure 2, how was the subset of samples shown selected?

c. Figure 3 and 4 are blurry. Please improve image resolution.

d. In Table 2, clarify if up/down references treated vs control or control vs treated.

e. There are no references to the supplemental figures in the text. Please add.

f. In line 70 and 76, the birds are listed as “Avian broilers”. Was this supposed to reference an Aviagen broiler

line? If so, please correct. If “Avian” is just referring to bird, remove this word.

g. In line 41-42, reference 5 is cited for using appendix, which is not a tissue in birds. Are they referring to the ceca?

h. In line 109, what does “not matched” mean? No hit to chicken genome?

i. The number of valid ESTs are reported in line 147 but how many of the failed clones were low quality sequence, unmatched sequence, repeat sequence, or unannotated protein sequence?

j. ITGA4 and DOCK10 should not be highlighted in the abstract and conclusions as these were not confirmed by qRT-PCR (p>0.05).

6. PLOS authors have the option to publish the peer review history of their article (what does this mean?). If published, this will include your full peer review and any attached files.

Reviewer #1: No

Reviewer #2: No

---

## [Author Response · Author response to Decision Letter 0]

29 Oct 2019

Dear academic editor and reviewers,

Thank you very much for giving us an opportunity to revise our manuscript. We appreciate the editor and reviewers very much for their constructive comments and suggestions on our manuscript entitled “Screening of differentially expressed immune-related genes from spleen of broilers fed with probiotic Bacillus cereus PAS38 based on suppression subtractive hybridization”.

We have studied reviewers’ comments carefully. According to the reviewers’ detailed suggestions, we have made a careful revision on the original manuscript. All revised portions are marked in red in the revised manuscript which we would like to submit for your kind consideration.

Kind regards.

Jiajun Li

E-mail: 18227585635@163.com

Corresponding author: Kangcheng Pan

E-mail: pankangcheng71@126.com

Reviewer #1:

Thank you for your sincere suggestion. I've replaced the blurred images with high-quality figures.

Reviewer #2:

1. Why was SSH selected as the method for predicting differentially expressed genes? Since the chicken genome is fairly well annotated, why was RNA-seq or 3’ Quant-Seq (better suited to large sample numbers) not used? These methods can be statistically tested (unlike SSH) and also screen the whole transcriptome without a priori knowledge of the treatment effect.

Firstly, a large number of previous studies have shown that SSH is a good method for constructing transcriptomes and screening differentially expressed genes. Secondly, this research is part of the National Natural Science Foundation (China) that our research group applied for in 2014. At that time, RNA-seq or 3'Quant-Seq technology was not mature. Third, although RNA-seq or 3’ Quant-Seq has been widely used in the determination of large Numbers of samples, it is inconvenient for the determination of a small number of samples. Furthermore, using different methods, the results of different genes can be different or the same. So, SSH was used to complete the experiment in our study.

2. Perhaps the low confirmation rate of the qRT-PCR reflects the lack of normalization to a reference gene. Even for absolute qRT-PCR, the copy number for each test gene needs to be normalized to a reference gene to account for variation in total input RNA. Please add the reference information if already done, or complete an additional qRT-PCR using a stably expressed reference gene, adjust the copy number for the 9 target genes and run the ANOVA again.

We have supplemented the absolute qRT-PCR for reference genes as required. The results showed that the expression of the reference gene β-actin was almost the same between the treated group and the control group, and there was no significant difference (P>0.05). At the same time, we also add the corresponding figures and data to the manuscript. In addition, in the process of revising the manuscript, we found that the calculation of plasmid standard concentration was wrong, so we used the Bio-Rad CFX Manager 3.1 software to reset the concentration of plasmid standard, rebuild the standard curve, and recalculate the copies of the differential genes. Then we compared the differences of the new calculation results again, and modified the corresponding discussion part and figures in the manuscript.

3. Please strengthen the evidence in the introduction for Bacillus cereus as a probiotic.

We have added the introduction of Bacillus cereus as a probiotic as required.

4. The discussion included very few references for chicken, with none for the significant JCHAIN, FTH1, and P2RX7 genes. Please add literature for these 3 genes in chicken.

We have added these three genes to the discussion in chicken as required.

5. The GO term results in lines 149-152 are not interpreted clearly. As analyzed, the biological process, molecular function and cell component terms are considered as if they are mutually exclusive and each EST can only have 1 type of term. It would be better to report this information as the percentage of the total ESTs in each category (so each EST can be represented 0-3 times) rather than a percentage of total GO terms.

We have reinterpreted the results as required.

6. Figure S6 and S8 show that IGF1R and ITGA4 primers have really low efficiency (60%). Were any other primer pairs tested for these genes? Perhaps an alternate would amplify better and better discriminate between the treated and control groups.

We have redesigned the primers, and re-tested the absolute qRT-PCR of these two genes and changed the relevant data in the article.

7. A gene set enrichment analysis (such as with PANTHER or GSEA) might be more informative than the top level GO terms from DAVID.

According to your suggestion, we use blast2go software to re-analyze the data and change the relevant content of the article.

8. Based on the feeding protocol, there were 30 control and 30 treated birds. How many spleens from each group were used to perform the SSH and were the samples pooled? How many individual samples/group were used in the qRT-PCR validation? Add to the text and to figure legends. Also note that if all 60 birds were used, the phrase “randomly selected” in line 81 doesn’t make sense.

We randomly selected two chickens from each repeat of each group, that is, a total of six chickens in each group. We extracted the total RNA of each spleen separately and aggregated the total RNA of each group.

We used the total RNA collected from the treated group and the control group for reverse transcription and a total cDNA was obtained respectively. And then we use this total cDNA for qRT-PCR validation.

We have supplemented the relevant content in the article as required.

9. There are many other experimental details missing from the methods section, including:

a. What is the “basic diet”? Please move Table S3 into the main manuscript, as the composition of the diet is important to the interpretation of this study.

We have moved the basic diet ingredient list to the manuscript as required.

b. Please put the information in Table S1 and S2 into the text of the methods and not supplemental tables, so that the company and reagent information is complete in the text.

We have moved the contents of two supplemental tables into the manuscript as required.

c. What type of housing (floor pens, cages, etc.) was used for the birds?

We have supplemented the notes in the manuscript as required.

d. Please add the bacterial growth conditions (media, temp, etc.), and how the bacteria concentration was determined to line 72-74.

We have added relevant content to the manuscript as required.

e. In line 87, “diluted 5 times” is not clear. Please clarify if this means a 5-fold (1:5) dilution or 5 serial dilutions to what final dilution factor.

We have clarified in the manuscript as required.

f. How much input RNA was used for cDNA synthesis in line 87 and 93-94? How much cDNA was used as input for the qRT-PCR reactions in line 116?

We have supplemented the manuscript as required.

g. In line 98, what were the hybridization conditions? Also add to this section, the enzyme and condition used for digestion, as mentioned in line 138.

We have added relevant content to the manuscript as required.

h. For all reactions (cDNA synthesis in line 87 and 93-94, nested PCR in line 98, liquid-culture treated PCR in line 102-103, and qRT-PCR in line 116), the components of the reactions and the cycling conditions (i.e. the program) need to be described. For the qRT-PCR, please also include a description of how the standard curves were generated/used and reference the supplemental figures.

We have added relevant content to the manuscript as required.

We have explained how the standard curve is generated/used in the manuscript as required, as well as the reference supplementary figures.

i. Were amplicons from the qRT-PCR confirmed by sequencing?

When we prepared the plasmid standard, we sent the plasmid standard to be sequenced and confirmed that there was no problem. The primers used for qRT-PCR are the same as those used for preparation of plasmid standard. We don't think it is necessary to send them for sequencing confirmation, so the amplicons from the qRT-PCR were not sequenced.

j. In line 110, what version of the DAVID database was used?

We used version 6.8 of DAVID before, but now we have changed it to blsat2go software.

k. What were the positive and negative controls for the SSH and qRT-PCR validation?

Based on the SSH test method and previous references, we believe that it does not require positive and negative controls.

The positive control of qRT-PCR was the standard plasmid diluted 10-fold, while the negative control was the system with RNase-free water as template.

l. The percentage agarose used for electrophoresis is not consistent. Is it 1.2% (line 90 and 128) or 1.5% (line 133 and 141)?

We apologized for the incorrect concentration of agarose gel electrophoresis in the manuscript. In fact, we used 1.2% concentration agarose gel in the experiment. We have corrected the relevant information in the manuscript.

10. Other comments:

a. There are English grammar errors in the manuscript, please check.

We have checked and corrected English grammar errors as much as possible according to the requirements.

b. In Figure 2, how was the subset of samples shown selected?

We randomly selected 400 white clones from plate Petri dishes for PCR. Then, the PCR products were electrophoretized and many electrophoretic results were obtained. We randomly selected some electrophoretic results for display.

c. Figure 3 and 4 are blurry. Please improve image resolution.

We use blast2go software to re-analyze the data and get new high-quality figures. We will re-upload the new figures.

d. In Table 2, clarify if up/down references treated vs control or control vs treated.

We have made it clear in the manuscript as required.

e. There are no references to the supplemental figures in the text. Please add.

We have mentioned the supplemental figures in the manuscript as required.

f. In line 70 and 76, the birds are listed as “Avian broilers”. Was this supposed to reference an Aviagen broiler line? If so, please correct. If “Avian” is just referring to bird, remove this word.

“Avian broilers” is an avian white feather broiler, we have made it clear in the manuscript as required.

g. In line 41-42, reference 5 is cited for using appendix, which is not a tissue in birds. Are they referring to the ceca?

I'm sorry we didn't read that reference carefully before. After careful reading, we found that "appendix" is their writing error, it should be "cecum", we have corrected the relevant content in the manuscript.

h. In line 109, what does “not matched” mean? No hit to chicken genome?

"Not matched" means that no sequence can be compared in NCBI databases. These sequences may be erroneous sequences that occur during sequencing or new gene sequences.

i. The number of valid ESTs are reported in line 147 but how many of the failed clones were low quality sequence, unmatched sequence, repeat sequence, or unannotated protein sequence?

We have added the result table of sequence alignment to the supplementary file.

j. ITGA4 and DOCK10 should not be highlighted in the abstract and conclusions as these were not confirmed by qRT-PCR (p>0.05).

We have deleted the content of these two genes in abstract and conclusions as required.

---

## [Decision Letter · Decision Letter 1]

9 Dec 2019

Screening of differentially expressed immune-related genes from spleen of broilers fed with probiotic Bacillus cereus PAS38 based on suppression subtractive hybridization

PONE-D-19-18103R1

Dear Dr. PAN,

We are pleased to inform you that your manuscript has been judged scientifically suitable for publication and will be formally accepted for publication once it complies with all outstanding technical requirements.

With kind regards,

Juan J Loor

Academic Editor

PLOS ONE

Additional Editor Comments (optional):

Reviewers' comments:

Reviewer's Responses to Questions

**Comments to the Author**

1. If the authors have adequately addressed your comments raised in a previous round of review and you feel that this manuscript is now acceptable for publication, you may indicate that here to bypass the “Comments to the Author” section, enter your conflict of interest statement in the “Confidential to Editor” section, and submit your "Accept" recommendation.

Reviewer #1: All comments have been addressed

Reviewer #2: All comments have been addressed

2. Is the manuscript technically sound, and do the data support the conclusions?

Reviewer #1: Yes

Reviewer #2: Yes

3. Has the statistical analysis been performed appropriately and rigorously? 

Reviewer #1: Yes

Reviewer #2: Yes

4. Have the authors made all data underlying the findings in their manuscript fully available?

Reviewer #1: Yes

Reviewer #2: Yes

5. Is the manuscript presented in an intelligible fashion and written in standard English?

Reviewer #1: Yes

Reviewer #2: Yes

6. Review Comments to the Author

Reviewer #1: The Authors have revised their paper according to the comments received, so in my opinion the revised paper merits the final acceptance.

Reviewer #2: Thank you to the authors for their detailed work addressing the comments in their response letter and revised manuscript.

Two tiny comments:

In the new Blast2Go pie charts, there are watermarks behind the images. The images would be clearer without these.

As the corrected qPCR analysis now confirmed (p-value<0.05) the down-regulation of ITGA4 and DOCK10, these genes can be put back in the abstract and conclusions (contrary to the previous suggestion to remove them).

7. PLOS authors have the option to publish the peer review history of their article (what does this mean?). If published, this will include your full peer review and any attached files.

Reviewer #1: No

Reviewer #2: No

---

## [Editor Report · Acceptance letter]

11 Dec 2019

PONE-D-19-18103R1 

Screening of differentially expressed immune-related genes from spleen of broilers fed with probiotic *Bacillus cereus* PAS38 based on suppression subtractive hybridization 

Dear Dr. PAN:

I am pleased to inform you that your manuscript has been deemed suitable for publication in PLOS ONE. Congratulations! Your manuscript is now with our production department. 

With kind regards,

on behalf of

Dr. Juan J Loor 

Academic Editor

PLOS ONE